# Assessing Genetic Algorithm-Based Docking Protocols for Prediction of Heparin Oligosaccharide Binding Geometries onto Proteins

**DOI:** 10.3390/biom13111633

**Published:** 2023-11-09

**Authors:** Samuel G. Holmes, Umesh R. Desai

**Affiliations:** 1Department of Medicinal Chemistry, School of Pharmacy, Virginia Commonwealth University, Richmond, VA 23298, USA; holmess@vcu.edu; 2Institute for Structural Biology, Drug Discovery and Development, Virginia Commonwealth University, 800 E. Leigh Street, Suite 212, Richmond, VA 23219, USA

**Keywords:** heparin/heparan sulfate, molecular docking, glycosaminoglycans, knowledge-based docking

## Abstract

Although molecular docking has evolved dramatically over the years, its application to glycosaminoglycans (GAGs) has remained challenging because of their intrinsic flexibility, highly anionic character and rather ill-defined site of binding on proteins. GAGs have been treated as either fully “rigid” or fully “flexible” in molecular docking. We reasoned that an intermediate semi-rigid docking (SRD) protocol may be better for the recapitulation of native heparin/heparan sulfate (Hp/HS) topologies. Herein, we study 18 Hp/HS–protein co-complexes containing chains from disaccharide to decasaccharide using genetic algorithm-based docking with rigid, semi-rigid, and flexible docking protocols. Our work reveals that rigid and semi-rigid protocols recapitulate native poses for longer chains (5→10 mers) significantly better than the flexible protocol, while 2→4-mer poses are better predicted using the semi-rigid approach. More importantly, the semi-rigid docking protocol is likely to perform better when no crystal structure information is available. We also present a new parameter for parsing selective versus non-selective GAG–protein systems, which relies on two computational parameters including consistency of binding (i.e., RMSD) and docking score (i.e., GOLD Score). The new semi-rigid protocol in combination with the new computational parameter is expected to be particularly useful in high-throughput screening of GAG sequences for identifying promising druggable targets as well as drug-like Hp/HS sequences.

## 1. Introduction

High-throughput molecular docking is an oft-used computational technique in drug discovery, which requires extensive sampling of bound conformations of each molecule in a library and concomitant scoring of the interactions with a putative site of binding on a protein. Popular software packages available for this purpose include AutoDock, MOE, DOCK, GOLD and others, which purport to efficiently sample the conformational space available to both the ligand and the protein in a short time. Unfortunately, this is achievable only if the number of rotatable bonds is not high [1,2,3], which is typically possible for small, drug-like hydrophobic molecules.

Molecular docking and scoring techniques have also been implemented for biomolecules such as smaller peptides and nucleic acids, which have garnered much interest as therapeutics in recent decades [4,5,6]. Specialized docking protocols have been designed to address their higher flexibility [7,8,9,10,11,12,13]. In fact, these docking protocols have advantageously utilized the vast amount of structural data available for peptides. For example, the ensemble docking of pre-generated conformer libraries based on experimentally determined structures have significantly improved docking efficiency [13,14]. These structural datasets have been leveraged in developing forcefield parameters for on-the-fly sampling of peptide conformations using physics based-approaches [12,15].

A super-class of arguably more flexible biopolymers—the glycosaminoglycans (GAGs)—is a family of four unique structures including hyaluronic acid, chondroitin/dermatan sulfate, heparin/heparan sulfate (Hp/HS), and keratan sulfate. These biopolymers are made up of repeating disaccharide building blocks in which a glucuronic acid (GlcA), iduronic acid (IdoA) or a galactose residue is connected to either a glucosamine (GlcN) or galactosamine residue through 1→3 or 1→4 inter-glycosidic linkages (Appendix A). Interestingly, except for HA, GAGs can be variably *N*- or *O*- sulfated and *N*-acetylated at various positions by a panel of biosynthetic enzymes in the Golgi, which generates an astounding level of configurational and conformational diversity. For example, nearly 139 billion theoretical topologies are possible for a hexamer of Hp/HS prepared from one of its 72 building blocks assuming occupancy of only two puckers (^1^C_4_ and ^2^S_O_, Appendix A) for its IdoA residues. By comparison, a hexapeptide or a hexanucleotide may occupy only about 64 million or 4000 topologies, respectively (Appendix A).

A major corollary of the topological diversity of GAGs is that these biomolecules recognize and bind to a huge number and diversity of proteins. The latest compilation pegs the GAG interactome to be 3464 strong [16]. Understanding such an interactome necessitates the use of in silico approaches, which have already been used in the past to elucidate the GAG-binding potential of proteins. Methods used so far include amino acid primary sequence analysis [17,18], surface electrostatics [19,20], and dynamic molecular dynamics [21,22]; a high-throughput computational approach that rapidly predicts the strength and binding geometry of each GAG sequence in the library of millions is critically needed to identify biologically relevant GAG–protein pairs. Unfortunately, such a computational approach is a pipe-dream at this time. Although dedicated software packages are available, including GlycoTorch Vina [23], Vina-Carb [24], and ClusPro with heparin extension [25,26], basic GAG docking is still challenging and limited.

One major limitation faced by GAG docking approaches is the paucity of highly efficient, and highly reliable, docking protocols. Currently, GAG docking protocols have implemented either “rigid” or “flexible” inter-glycosidic linkages (Φ and Ψ) to predict the strength and binding geometry of sequences. To a large extent, these approaches have been used to predict the sites of GAG binding, the length and fine structure of the preferred GAG sequence(s) and the native GAG binding pose. A high-throughput protocol put forward in recent times is the combinatorial virtual library screening (CVLS) protocol, which has achieved considerable success in the screening of a library of GAG sequences against different proteins such as antithrombin, spike glycoprotein, heparin cofactor II, transforming growth factor beta, and insulin-like growth factor–1 receptor [27,28,29,30,31]. CVLS employs rigid Φ and Ψ values, which are taken from the literature on related sequences and fixed at the midpoint of the known range. Many other groups, including those of Samsonov, Gandhi, Ricard-Blum, Rusnati, and Pisabarro, have also implemented rigid docking on a number of GAG sequences to much success, including bone morphogenetic protein 2, αvβ3 integrin, HIV-1 matrix protein p17, CXCL12 [32,33,34,35] and many others.

By the same token, flexible docking has also helped shed light on the GAG recognition of proteins, especially Hp and CS [35,36,37]. Interesting insights can be gained through flexible docking, as shown in a CCL5 study with a small library of Hp tetrasaccharides that revealed a novel binding mechanism involving the protonated His23, which is populated at pHs less than 7 [37]. Unfortunately, flexible docking generally succeeds for smaller GAG chains [37,38]. Sometimes molecular dynamics is used as a follow up method to “clean-up” top-ranked poses of flexible docking, which significantly restricts the application of this approach under a high-throughput format [36,39,40]. To address this, coarse grain (CG) modeling with full flexibility has been implemented for longer GAG oligosaccharides; yet, even here the CG beads attempt to drastically reduce the flexibility arising from ring substituents [41].

We reasoned that fundamental, comparative studies on rigid and flexible docking protocols are necessary to understand their applicability for the high-throughput screening of a large library of GAG sequences. We also reasoned that an alternate protocol, i.e., semi-rigid docking, be assessed before embarking on designing a robust, high-throughput algorithm. In this work, we report a comparative study of docking 18 Hp/HS oligosaccharides, ranging from di- to decasaccharide, onto their targets using “rigid”, “flexible” and “semi-rigid” protocols. We have analyzed their successes in locating the “native” pose, as observed in the co-crystal structures. We have chosen a structurally diverse range of proteins exhibiting a range of selectivities—from the highly selective (antithrombin) to highly non-selective (thrombin). We find that although all three protocols succeed reasonably well, the rigid and semi-rigid protocols recapitulate crystal structure poses for chains as large as a decasaccharide more often and in a reproducible manner. Our work shows that for an unknown system, the semi-rigid protocol is a better alternative to employ in comparison to the rigid (when the “native” Φ/Ψ are not known). Likewise, the semi-rigid docking protocol is much better than the flexible docking protocol, especially for longer GAGs (5→10 mers) and also for smaller sequences (2→4 mers). Finally, we present a computational parameter that could be employed in the virtual high-throughput screening of thousands of sequences for identifying putative drug-like Hp/HS structures.

## 2. Methods

### 2.1. Software

SYBYLX 1.3 (Tripos Associates, St. Louis, MO, USA) was used for molecular visualization, minimization, and protein/ligand preparation. GOLD Version 2020 was used for molecular docking experiments [1].

### 2.2. Structure Selection and Preparation

Co-crystal structures were generally selected based on the reported resolution (≤2.5 Å) and containing the entire structure of the sequence used for crystallization (2→10-mer) (Appendix A). For 5DNF, 2VRA, 3EVJ, 7B8I, and 1E0O, no oligomer structures were available that met the resolution cut-off and hence the next best co-crystal structure was utilized. Unnecessary subunits, cofactors, and metal ions that did not directly bind to the ligand were removed. The Hp/HS sequence was extracted followed by the removal of water molecules. Hydrogens were added to the protein in SYBYL X1.3 and the protein was minimized with fixed heavy atom coordinates using the Tripos force field for 100,000 iterations followed by a termination gradient of 0.05 kcal/mol. Gasteiger–Hückel charges were introduced and structure minimization was carried out with a non-bonded interaction cut-off of 8 Å and a dielectric constant of 80.

### 2.3. Hp/HS Sequence Preparation

Post extraction of the sequence, structures were manually inspected to ensure correct atom and bond typing. The Tripos force field does not contain parameters for sulfate oxygens, so the carboxylate oxygen (O.CO_2_) atom type was used as a surrogate, as documented in earlier studies for the parameterization of GAGs [30,42]. Hydrogen atoms were added to the ligand in SYBYLX 1.3.

### 2.4. Molecular Docking

The default ligand binding site definition in GOLD was used for all studies in this work. In this definition, all residues that are within 6 Å of the native pose are selected. The protein was kept rigid during the docking, while the Hp/HS ligand was allowed varying degrees of flexibility depending on the protocol being used, except for ring torsions (see Figure 1A). Glycosidic torsions from each crystal structure were used as reference for rigid and semi-rigid dockings. In the rigid docking protocol, the glycosidic torsions were restricted to the initial values, while all other non-terminal single bonds were allowed free rotation. In contrast, flexible docking allows free rotation at glycosidic linkages. For semi-rigid dockings, the user specifies torsional histograms for Φ and Ψ for each glycosidic linkage, where the center of the bell curve corresponds to the initial value. Although we chose a 30° cutoff in this study, the user can adjust these values if needed. To generate a torsional histogram in GOLD, we refer the reader to the GOLD user guide, but we provide a brief example in the Appendix A. For dockings, 100 GA and 300 GA runs were used with 10,000 genetic operations each. Each docking experiment was performed in triplicate and the two poses from each run with the highest GOLD Scores were retained for analysis. A total of six poses per structure were utilized for analysis of docking protocol performance. Pairwise RMSDs and glycosidic torsions were calculated via in-house python scripts. All molecular models were generated in PyMOL.

## 3. Results

### 3.1. Basis Underlying the Semi-Rigid Docking Protocol

Although GAGs contain multiple ring substituents (–H, –OSO_3_^−^, –NHCOCH_3_, –NHSO_3_^−^) that are flexible, glycosidic torsions (Φ/Ψ) and ring puckering (^1^C_4_/^2^S_O_) are the bane of conformational challenges in docking studies. A typical rigid docking protocol assumes that glycosidic torsions and ring puckers are restricted to their initial values during the search, while imparting flexibility to ring substituents (Figure 1A). In contrast, a fully flexible docking algorithm imposes no constraints on glycosidic torsions and ring substituents. Whereas the rigid docking protocol samples minimal conformational space, the flexible protocol attempts to sample the entire conformational space. Alternatively, the two protocols represent two extreme ends.

It has now been long known that GAG glycosidic torsions in oligosaccharides sample a rather limited range of ~60°. Curating Φ/Ψ values from the PDB shows two dominant clusters corresponding to two categories (GlcN→UA and UA→GlcN) with only a few unusual torsions (Figure 1B, Appendix A). In fact, these torsional preferences were exploited in developing VinaCarb and GlycoTorch Vina (GTV), two computational tools in which low-energy ligand poses are pre-generated using quantum mechanical (QM) energy wells predicted by glycosidic torsion’s specific carbohydrate intrinsic (CHI) energy functions [23,43].

With regard to ring puckers, the flexibility of IdoA is thought to be a key contributor to selective recognition [44,45,46]. IdoA can sample a number of ring puckers; however, ^1^C_4_ and ^2^S_O_ have been shown to be the most populated with their dynamic equilibrium determined by the neighboring sulfation pattern [47]. Several authors have explored the role of IdoA puckers in protein recognition [44,48,49]; however, Boittier et al. have used GTV to elucidate the effects of IdoA puckers (especially ^2^S_O_) on the internal energy of the GAG chain [23]. Interestingly, redocking with recalculated CHIs for seven common GAG linkages using density functional theory resulted in the recapitulation of 5/12 test cases. Unfortunately, GTV accuracy appears to taper off for chains longer than a hexasaccharide.

Given the current state of knowledge and challenges, we reasoned that a simple, rapidly implementable, reasonably comprehensive approach to recapitulate native poses of GAG oligosaccharides would be to rely on torsional probability distributions that are derived from Φ/Ψ averages. As shown in Figure 1B, low-energy torsions are found to populate a rather narrow range of ±30° (see also Appendix A for all compiled data). Such a probability distribution would afford the flexibility necessary for an optimal fit into the binding pocket, which may be bypassed during a purely rigid docking search. At the same time, the partial flexibility of ±30° would also avoid the massive, and unnecessary, conformational search that accompanies a fully flexible docking search. We hypothesized that such a probabilistic bias of ±30° around the common Φ/Ψ average could increase the likelihood of finding the native pose. We call this approach the “semi-rigid” docking (SRD) protocol. However, before implementing such a protocol for high-throughput purposes, key questions need to be addressed, including (i) how useful are rigid and flexible docking approaches? (ii) Can the SRD protocol better for longer GAG chain? and (iii) does it recapitulate the native pose of a GAG–protein complex? This work addresses these and other comparative questions.

### 3.2. Does Rigid Docking Recapitulate the Native Pose?

Although ligand geometries presented in the PDB entries may not necessarily be accurate, the large majority are assumed to be the “native”, most preferred, binding geometries [50]. It is expected that as a ligand–protein complex crystallizes, the “native” pose outcompetes other possible poses because of stereo-electronic restrictions imposed by a well-defined binding site. Unfortunately, most sulfated oligosaccharides bind in a shallow, surface-exposed, flexible, highly water-laden, cationic binding site [20,26,51]. This raises a question as to whether the “selectivity” engineered by crystallization forces represents an intrinsic property of the GAG–protein complex. In fact, the crystal structure of a hedgehog–heparan sulfate complex exhibited dimerization by binding two different, adjacent HS sequences on the same chain with different conformations, despite interacting with the same five residues [52]. Likewise, the prototypic, non-selective, GAG-binder thrombin displays at least two crystal poses for Hp/HS oligomer that differ significantly in torsional and translational profiles (Appendix A). Given this ambiguity, it is useful to assess whether the “native” crystal structure pose is recapitulated upon docking, especially when the GAG is conformationally immobile during docking. As stated above, “rigid” docking holds Φ/Ψ and ring puckers invariant during docking and equal to their crystal structure values, while affording flexibility to ring substituents (Figure 1A).

We studied the rigid docking protocol on 18 proteins that had been co-crystallized with an oligosaccharide using GOLD, a genetic algorithm (GA)-based protocol. The Hp/HS oligosaccharides ranged in length from di- to decasaccharide. Their fine structure could be classified into primarily two types including those containing the common repeating sequence and some containing the rare 3-*O*-sulfated sequence (Appendix A). Our docking protocol implemented 100 GA runs with 100,000 operations per run, which has been employed in numerous studies in the literature [53,54] and was found to be very useful for GAGs also [20,30,42]. Experiments were performed in triplicate for each Hp/HS oligosaccharide, and the top two poses each 100 GA run, i.e., six poses in toto, were collected. Analysis of the collected poses was performed in a quantitative manner through the use of a parameter called RMSD, which stands for root mean square difference, between either the native and predicted poses or between different predicted poses. In the literature, an RMSD of 2.5 Å or less has been deemed as geometric equivalence between two poses [20,55,56]. For most part, the cut-off of 2.5 Å typically works for small molecules and smaller oligosaccharides (<6 mers). It is important to recognize that longer oligosaccharides may not adhere to this arbitrary cut-off. Alternatively, RMSD cut-off may have to be scaled with chain length. Yet, we find that the RMSD cut-off of 2.5 Å worked for several oligosaccharides longer than 6 mers (see below).

We performed three comparisons of the docked pose(s) with the native pose observed in the co-crystal structures (Figure 2A). (i) We first calculated RMSD_AVERAGE_ as the difference between the native pose and the average of the top six rigid docking poses. (ii) Then, RMSD_LOWEST_ was calculated to quantitatively assess the difference between the native pose and the one docked pose that most closely matches the native. This parameter has been an oft-used method reported in the literature [57,58,59]. (iii) We finally calculated RMSD_INTRAPOSE_, which assessed the similarity between the six docked poses. Calculation of these three RMSDs afforded a rather clear quantitative insight into the similarity of recognition, as evident from two representative sequences—a tetrasaccharide 6LJL and a disaccharide 1U4L—which present successful recapitulation (RMSD ≤ 2.5 Å) of the native pose and a not-so-successful prediction (RMSD > 2.5 Å), respectively (Figure 2B). 

Appendix A presents the comparative overlays of docked poses using rigid docking with the native pose for all 18 sequences. As evident from visual inspection, and counter to the simplistic expectation, rigid docking more frequently does not recapitulate the native pose found in the PDB co-complex. In fact, RMSD_AVERAGE_ calculation shows that only 6 of 18 HS oligosaccharides (33%) recapitulated the native pose (Figure 2C). Even when only the best pose is compared, as in RMSD_LOWEST_, recapitulation increases to 44% (Figure 2E). Curiously, none of the di- or trisaccharides recapitulated their native pose, whereas one each of tetra-, octa- and decasaccharide and two of the penta- and hexasaccharides recapitulated the native poses. This is interesting because the di- and trisaccharides are theoretically expected to span much smaller conformational space than the longer oligomers. 

When RMSD_INTRAPOSE_ is calculated, 50% of sequences were found to bind very consistently (RMSD ≤ 2.5 Å) (Figure 2D). Of these, 33% recapitulate the native pose (Figure 2C), which implies that three sequences (1U4L(2), 1U4M(2), and 4C4N(6)) bind consistently away from the site of the native ligand. Alternatively, these oligosaccharides converged to a single pose, but this was significantly different from the native pose. We also performed docking experiments using 3-fold-higher GA runs (i.e., 300 GA runs; each with 100,000 genetic operations) to assess whether more sampling helps the recapitulation of the native form; however, this did not yield any significant improvement in recapitulation success (Appendix A). This suggests that our genetic algorithm-based approach is not able to identify the “native” pose in the rigid docking paradigm, especially for 2→4 mers.

### 3.3. How Does Flexible Docking Approach Compare to the Rigid Approach?

Next, we evaluated the impact of imparting full flexibility to all rotatable bonds in an Hp/HS chain. Although GOLD holds pyranose ring puckers invariant from initial assignment, a typical Hp/HS sequence still encompasses at least five and seven rotatable bonds for all the UA and GlcN residues, respectively. This means that flexible docking minimally requires searching conformational space for 12 to 60 rotatable bonds for di- to decasaccharide, respectively (Figure 3A). In fact, the time for a triplicate flexible docking run steadily increases with chain length despite the limitation of 100,000 iterations (Figure 3B). Yet, this may not lead to the effective recapitulation of the native pose; rather, as the chain length increases, one may predict that flexible docking will not effectively recapitulate the native pose.

Figure 3C–E show RMSD_AVERAGE_, RMSD_INTRAPOSE_ and RMSD_LOWEST_ profiles for the 18 proteins at 100 GA runs. RMSDs were within the 2.5 Å threshold for 2, 3 and 5 Hp/HS chains out of 18 when “average”, “intrapose” and “lowest”, respectively, were compared to the native pose. As predicted, these successes are much lower than the corresponding results for rigid docking (six, eight and nine, respectively). Increasing the number of GA runs to 300 does not yield any significant improvement (Appendix A). Yet, some very interesting results can be gleaned from flexible dockings.

One, the flexible docking of two HS oligosaccharides appears to consistently converge to the native pose. These include a pentasaccharide (1TB6), which is universally recognized as the prototypic high-affinity, high-specificity antithrombin-binding Hp/HS oligosaccharide [60], and a tetrasaccharide cleavage product (6LJL), which is generated by a unique exolytic heparinase [61] with a high level of substrate specificity. Thus, flexible docking appears to work primarily for sequences that exhibit a high level of selectivity.

Two, if only the most native-like pose of top six docked poses is taken, as in RMSD_LOWEST_, the performance improves to 5 out of 18 chains (Figure 3E). Interestingly, these five sequences are tetrasaccharide or longer. Alternatively, none of the di- and trisaccharides recapitulated the native pose, results similar to those in the rigid docking study. In fact, a correspondence between rigid and flexible dockings can be noted for longer HS chains also.

Three, when RMSD_INTRAPOSE_ values are considered, only three HS chains pass the threshold, including 3B9F, 6LJL, and 1TB6. This result is dramatically different from that of the rigid docking protocol, wherein 44% of HS oligosaccharides converged to a single pose. This result reiterates that the flexible docking protocol tends to yield a wider array of significantly different poses, which also turn out to be different from the native pose. Alternatively, the flexible docking of GAGs appears to be inherently beset with a huge conformational space that is difficult to sample comprehensively within a reasonably short time.

### 3.4. Can Semi-Rigid Docking Approach Offer a Better Alternative?

To assess whether a rational, balanced docking approach can offer a better alternative, we studied a semi-rigid docking (SRD) protocol. As described in the section on rationale above, this protocol relies on the well-established understanding that glycosidic torsions Φ and Ψ prefer a rather narrow range of ±30°, irrespective of the local structural heterogeneity (Figure 1B, Appendix A). More specifically, it is usually Ψ, rather than Φ, that tends to vary more [23,62,63]. Further, two fairly defined Φ and Ψ minima are generally observed for the more flexible UA→GlcN torsions (~−80°/−100° and ~−80°/65°), whereas only one Φ and Ψ minimum is typically populated for GlcN→UA torsions (~80°/−145°) [62] (Figure 1B). Thus, a better docking approach would be to pre-generate torsional probability distributions around the well-known Φ and Ψ, then harness the power of GA to enhance the probability of finding the native pose. Although the current approach utilizes Φ and Ψ torsions derived from crystallographic and/or NMR studies, in principle, other computational calculations such as DFT, QM, and/or MM [64,65,66,67] may also be used. Briefly, our SRD protocol utilized a torsional probability distribution function (Appendix A) around pre-chosen Φ and Ψ torsions corresponding to the typical minima observed in solution. GOLD docking and analysis was performed as described for the rigid and flexible docking methods above.

Figure 4 presents the RMSD_AVERAGE_, RMSD_LOWEST_ and RMSD_INTRAPOSE_ analyses following SRD re-dockings using 100 GA runs. This protocol demonstrated accuracies similar to that of the rigid protocol, with rates of 33%, 44%, and 55%, respectively, which did not change even at 300 GA runs (see Appendix A). Interestingly, the SRD was also unable to recapitulate the native pose for smaller chains, i.e., all di-/tri- and most tetra- saccharides (Figure 4A). Yet, SRD was much better than the flexible protocol, especially for sequences longer than pentasaccharide. In fact, for longer chains (≥5 mers), RMSD_AVERAGE_ exhibits a success rate of 45% for the SRD protocol in comparison to 9% for the flexible approach (Figure 3C and Figure 4A). More importantly, SRD was able to predict the crystal structure poses of longer sequences very well, as exemplified by 3UAN and 1E0O (Figure 4D).

The RMSD_INTRAPOSE_ analysis of the SRD results reveals some interesting insights. Four of six smaller chains (2→4-mers) demonstrate highly consistent recognition, albeit different from the native pose (Figure 4B). This proportion is nearly two-fold higher than that observed for rigid docking (71% vs. 43%), suggesting that limited flexibility is important for convergence. In fact, the SRD approach is more effective in identifying proteins that exhibit significant levels of consistent recognition in comparison to rigid and flexible protocols. For example, trisaccharide 5DNF displayed RMSD_INTRAPOSE_ of 1.28 Å for SRD in comparison to 3.95 Å and 4.02 Å for rigid and flexible protocols, respectively. Similar results were observed for the 3B9F and 2HYV sequences.

To gain more insight into variances from consistent recognition, we calculated the deviation of each torsion from the native pose following SRD and flexible docking. Figure 5 presents the results for two octa- and one decasaccharide, whereas Appendix A presents the results for all 18 chains. Although both φ and ψ deviated more for the flexible protocol in comparison to the semi-rigid, ψ exhibited huge deviations (Figure 5C,D), which appears to be the foundational reason for the lack of recapitulation of the native pose. When the differences are averaged across all glycosidic linkages, the flexible protocol consistently predicts deviations in excess of 20° (Figure 5E,F). Interestingly, the deviations in φ increase with the length of the chain, while those for ψ remain high essentially independently of chain length. These results convey the importance of affording only limited flexibility, if any, to both glycosidic torsions, especially ψ, during docking operations for better success in native pose predictions. 

### 3.5. The Enigma of Disaccharides Finding the Native Pose?

A priori, docking a disaccharide onto a pre-determined site of binding should be the easiest and most likely to succeed because of its small size, smaller conformational search space, etc. The literature reports many studies on Hp/HS disaccharides binding to different proteins [68,69,70,71] Yet, none of these studies appear to have studied docking consistency with the co-crystal structure. In this connection, we were very surprised with the results for disaccharides observed in this study. None of our three protocols succeeded to any extent. In fact, the RMSD_AVERAGE_ across the library of sequences ranged from 5 to 8 Å, which was several-fold higher than the threshold (≤2.5 Å) set for the assessment of consistency. Figure 6 shows a comparison of the poses observed following rigid, SRD and flexible docking to the native, co-crystal structure pose for all three disaccharides studied (see Appendix A for other 15 sequences). All three disaccharides, and especially the 3B9F sequence, prefer to bind in a pocket adjacent to the native binding pocket. Thus, our results raise two possibilities: (i) either crystallization induces smaller sulfated glycans into non-native poses, or (ii) the GA-based docking algorithm fails miserably because the in silico affinity of smaller glycans is not high.

To assess the two scenarios, we first reviewed the RMSD_INTRAPOSE_ calculated for the three disaccharides. For rigid and SRD protocols, these RMSDs were noted to be within the set threshold (≤2.5 Å), suggesting that the three disaccharides appear to recognize their target proteins consistently (Figure 2D and Figure 4B), albeit in non-native sites. Even for the flexible docking protocol, RMSD_INTRAPOSE_ was found to be very consistent for one of the three disaccharides (Figure 3D). Over the past decade or so, we have demonstrated that a high level of docking consistency, i.e., RMSD ≤ 2.5 Å, correlates well with a high level of selectivity of protein recognition [20,30,42,49]. Thus, it is very likely that the three disaccharides demonstrate non-native binding poses in the co-crystal form, while in solution they are likely to engage an altered site. Another support for this hypothesis arises from the GOLD Score, a measure of in silico affinity (Figure 7A). The three disaccharides demonstrate a reasonably high GOLD Score, which implies that the second possibility presented above is unlikely to be true. Thus, our studies raise an alert for both biologists and molecular modelers to be particularly careful in interpreting experiments on smaller sulfated glycans. Rather, our studies point to the importance of simultaneously performing both crystallography and computational experiments, especially on smaller sulfated glycans.

### 3.6. A Parameter for Identifying Putative Drug-Like GAG Sequences

A key question stymieing efforts on discovering drug-like GAGs is identifying protein targets that bind sulfated sequences in a highly selective manner. As stated above, we have previously used RMSD as a measure of selectivity for parsing a library of thousands of sequences into selective and non-selective ones [20,27,42]. Do any of these exhibit drug-like characteristics? This work affords a unique opportunity to parse the 18 Hp/HS sequences into putative drug-like sequences. A re-review of results (Figure 2C and Figure 4A) for rigid and SRD docking protocols reveals that one tetra- (6LJL), one penta- (1TB6), two hexa- (4AK2 & 3UAN), one octa- (3INA) and one decasaccharide (1E0O) exhibit an RMSD_AVERAGE_ less than the pre-set threshold for selectivity (≤2.5 Å). On the other hand, the flexible docking protocol predicts only two high-selectivity sequences (6LJL & 1TB6) (Figure 3C). Yet, for drug-like properties, high docking scores are critical because these serve as surrogates for solution affinity. Unfortunately, the non-selective recognition of GAGs can also yield high docking scores, as shown by the disaccharides (Figure 6A). We reasoned that the ratio of GOLD Score to RMSD would emphasize both high-affinity and high-selectivity, and thereby better identify drug-like sequences from the rest. This new parameter (GOLDScore/RMSD) was found to vary from a low of ~7 to a high of ~200 across the three docking protocols (Figure 7B). Interestingly, the most selective six sequences identified by RMSD analysis also stood out above the others, especially for rigid and SRD protocols. In a quantitative manner, the GOLDScore/RMSD parameter appears to have a clear threshold of 50 for segregating selective versus non-selective systems (dotted line in Figure 7B). Such a quantitative threshold, if supported in future experiments, is likely to provide excellent insight into the GAG recognition of proteins. 

## 4. Discussion

Studying GAG binding to proteins has remained challenging for multiple reasons, of which the difficult synthesis of a library of homogeneous sequences [72,73] and the lack of rigorous computational tools for high-throughput studies [41] are the primary barriers. In the latter case, the two major hurdles have been the large number of rotatable bonds within a relatively small structural frame and the shallow, surface-exposed, cationic binding sites on proteins. A good number of efforts have been directed to resolve these challenges including hybrid fragment-based/coarse grain docking [41], QM restraints on glycosidic torsions [23,43], and molecular dynamics methods [21]. In this context, our current work attempts to show that a knowledge-driven algorithm based on pre-generated low-energy glycosidic torsions offers an attractive alternative for most chains, especially those that are longer. Following a rigorous comparative study on a structurally diverse group of proteins exhibiting diversity of GAG recognition selectivities, we conclude the rigid and semi-rigid protocols recapitulate crystal structure poses for longer chains (5→10 mers) more often. Between the two protocols, differences observed for smaller sequences convey the special value of the SRD approach over the rigid docking approach. More importantly, for systems lacking co-crystal structure information (i.e., “native” Φ/Ψ not known), the SRD protocol is a safer alternative to employ over the rigid protocol. Likewise, the SRD is a much better approach than flexible docking for all oligosaccharide sequences, but especially for longer GAGs (5→10 mers). 

Our work has revealed some interesting results and insights into the different docking protocols and GAG recognition. One striking finding was that smaller oligosaccharides appear to bind to sites other than the crystal structure-determined binding sites, irrespective of the protocol used. Most probably, this arises from crystallization artifacts and will have to be followed up in rigorous computational and solution-based validation experiments.

A key observation of this work was that several co-crystal poses simply could not be recapitulated in any of the three docking approaches. These include RANTES, thrombin, platelet factor 4 (PF4), annexin A2, robo, and hedgehog. While it is difficult to ascribe the exact underpinnings of this result, several possibilities can be postulated. First, the algorithm used in all three protocols, i.e., the GA-based approach with an artificially defined limit of termination, is not sufficient to sample the entire conformational space of longer chain oligosaccharides, i.e., hexasaccharides. Second, the energy terms of the docking program (GOLD) may be better suited for certain types of interactions, e.g., hydrophobic, and not for others, e.g., ionic. Third, our algorithm excluded the effects of hydration and protein flexibility to cut down on computational costs, despite their well-established importance in ligand binding [74,75,76]. Fourth, we used Gasteiger–Hückel partial charges and a particular force field (Tripos) to parameterize each sequence. These charges and forcefields are estimations at best and may carry inaccuracies in energy calculations of GAG–protein systems, as any other parameterization protocols would carry. It is likely that GAG-specific parameters would be more suitable. Finally, the nature of Hp/HS binding to these proteins is likely to be predominantly non-selective, which could result in many equivalent, but non-identical, binding poses [46]. In fact, RANTES [37,77,78], thrombin [79,80,81], PF4 [82,83,84,85], annexin A2 [86,87,88], robo [89,90,91,92], and hedgehog [52,93,94] appear to not bind to a specific Hp/HS motif (Appendix A). In fact, RANTES and thrombin have been shown to bind GAGs other than Hp/HS [95,96,97]. Of all the above explanations, the final explanation presents a significant new insight into the GAG recognition of proteins with possible biological consequences. Yet, much work is needed on all the above explanations to gain the confidence regarding the new insight on the GAG recognition of proteins.

To our knowledge, this work on the SRD protocol is the first study of its kind for GAGs, although it has been used in the fields of proteins/peptides, nucleic acid docking and the de novo design of protein binders [8,98,99,100]. Glycosidic Φ/Ψ minima can be availed from NMR, crystallographic, and/or QM/MM methods. In this work, we have primarily relied on crystallographic reports from the protein data bank (PDB), which are not very structurally diverse. Computational experiments have documented variations in preferred Φ/Ψ for some 3-*O*-sulfated sequences, i.e., compact geometries [62]. It would be important to study and validate the applicability of the SRD protocol for such alternative geometries too. 

In this study, we selected GOLD as the docking platform and utilized Gasteiger–Hückel charges, because together they have been successfully used in the past in identifying selective sequences, which were validated through biophysical experiments [28,30,101]. Yet, the performance of GOLD has not been fully investigated in a comparative manner for GAGs, as done for other docking programs [102,103]. Thus, a comparative study of different parameters and docking programs in the application of the SRD protocol for the prediction of GAG binding geometries should be performed in the future. Likewise, it would also be important to test the applicability of the SRD protocol using GAG-specific parameters, including partial charges, which are absent in the current protocol [26,104]. We also emphasize that protein flexibility is currently neglected in the SRD protocol, and is likely to be important in GAG binding. Protein flexibility is often neglected in the virtual screening of GAGs to make the protocol computationally tractable. However, a two-step approach could be envisaged wherein the protein is held rigid in the first step to identify high-scoring sequences, which are then studied in the second step with binding site flexibility. 

A specific point related to the use of other docking platforms is whether multiple top geometries are accessible in the tool (and not necessarily only the topmost geometry). Here, each docking platform inherently calculates such geometries, which our work attempts to convey are important to analyze in a logical manner for specificity analysis. Thus, the principles enunciated here should be applicable irrespective of the docking platform used for the prediction of GAG, peptide, or drug complexes with proteins. Finally, the fundamental work continuing in the direction of developing better force-field parameters for GAGs, e.g., GLYCAM and others [46,47] will help build a better SRD protocol in the future.

This work presents a very useful parameter—GOLDScore/RMSD—that can be used to parse selective versus non-selective GAG–protein systems. In this work, we utilized RMSD_AVERAGE_ to identify some highly selective GAG–protein systems. Yet, it is important to speculate whether RMSD_AVERAGE_ or RMSD_INTRAPOSE_ should be used. We used RMSD_AVERAGE_ because co-crystal structures were available in the PDB. However, a large majority of GAG–protein complexes have not been crystallized. Thus, computationally derived RMSD_INTRAPOSE_ will be the only option for exploratory studies. While it may appear that the lower values of RMSD_INTRAPOSE_ may help identify more selective sequences, it is important to note that GOLD Score is also an important determinant in recognition and has to be reasonably high for the suggested threshold of 50. However, more work is needed to assess the validity of this parameter and the suggested threshold. Yet, this simple parameter is likely to find major value in virtual high-throughput screening studies for the rapid identification of putative drug-like sequences against targets of interest. 

## 5. Conclusions

This work presents foundational experimentation on developing a protocol for use in the high-throughput screening of Hp/HS oligosaccharides for protein recognition. Protein binding sites generally accommodate Hp/HS chains that are 4- to 8 mers in length [28,101,105,106]. The majority of proteins have not been crystallized with GAG partners. Thus, “native” Φ/Ψ remain unknown. Thus, for de novo applications, SRD seems to be more suitable. In this connection, the ratio parameter based on GOLD Score and RMSD values, which could be easily implemented in the virtual high-throughput screening of thousands of sequences, would be particularly useful in identifying putative drug-like Hp/HS structures. 

## Figures and Tables

**Figure 1 biomolecules-13-01633-f001:**
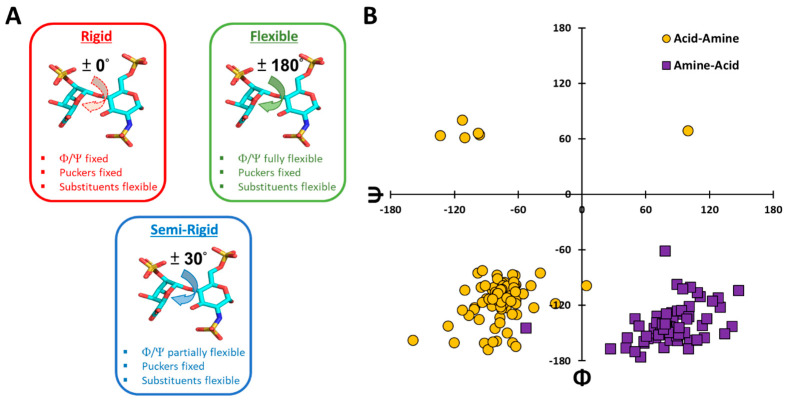
Types of docking protocols used in the literature. (**A**) Two types of protocols typically used for GAGs in the literature including “rigid” and “flexible” docking. This work reports comparative studies of these two with a semi-rigid docking protocol, which affords better predicting success, especially for longer GAG sequences. Whereas the rigid and flexible docking approaches hold glycosidic torsions (Φ and Ψ) either completely invariant (±0°) or fully flexible (±180°), the semi-rigid protocol allows partial flexibility (±30°) around the most preferred torsions reported in the literature. (**B**) A Ramachandran plot depicting Φ and Ψ for all 41 Hp/HS–protein complexes available in the Protein Data Bank (www.rcsb.org, accessed on 1 July 2021). Φ and Ψ are categorized into two groups: acid–amine (UA→GlcN) and amine–acid (GlcN→UA).

**Figure 2 biomolecules-13-01633-f002:**
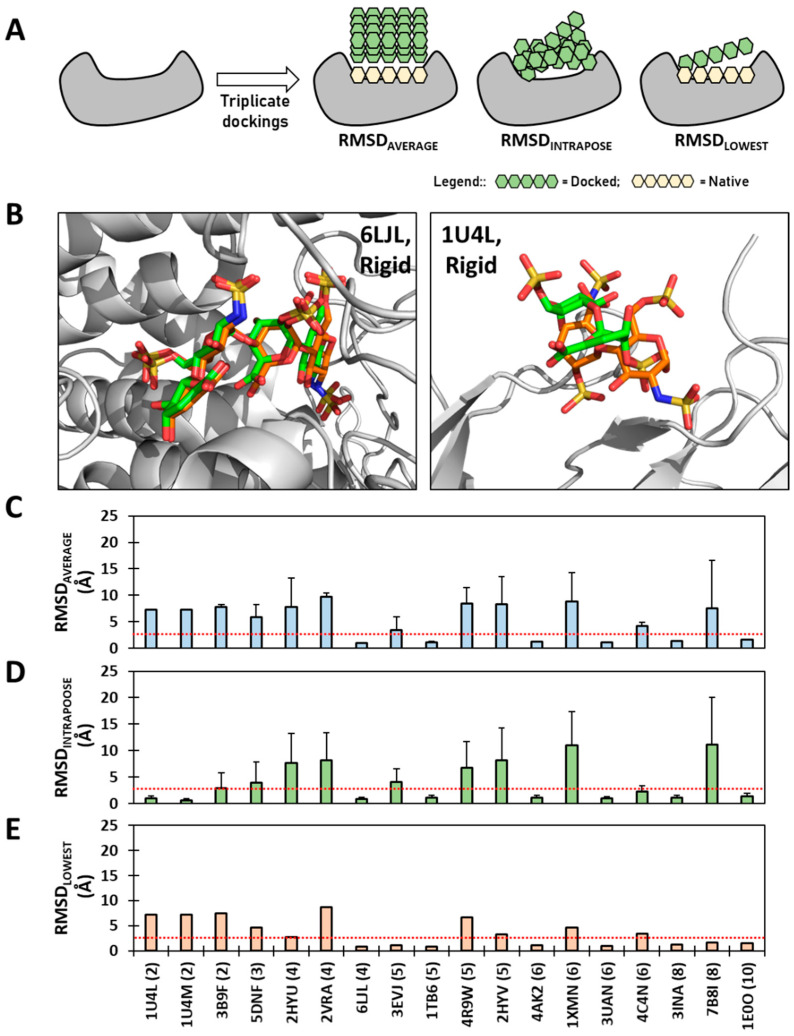
Recapitulation of the native pose using a rigid docking protocol. Each sequence was redocked back into the crystal structure in triplicate using 100 GA runs, each being allowed 100,000 genetic operations. The top two poses from each replicate experiment were selected, compiled and used for analysis. (**A**) The docking of each Hp/HS oligosaccharide onto its target protein was analyzed by calculating the RMSD_AVERAGE_, RMSD_LOWEST_ and RMSD_INTRAPOSE_, which convey the root mean square difference (RMSD) between the native pose and the average of the top six rigid docking poses, the difference between the native pose and the one docked pose that most closely matches the native, and the intra-pose difference between the six docked poses, respectively. (**B**) Representative example of a successful recapitulation (RMSD_AVERAGE_ ≤ 2.5 Å) of the native pose of an Hp/HS tetrasaccharide (left; 6LJL) and a not-so-successful predication (RMSD_AVERAGE_ > 2.5 Å) of the native pose an Hp/HS disaccharide (right; 1U4L). Native poses in both are shown in green, while docked poses are in orange. (**C**–**E**) Three different RMSDs as function IDs of the co-complex structures reported in the PDB. *X*-axis labels represent the PDB code followed by chain length in brackets. The red dotted line indicates the 2.5 Å cut-off.

**Figure 3 biomolecules-13-01633-f003:**
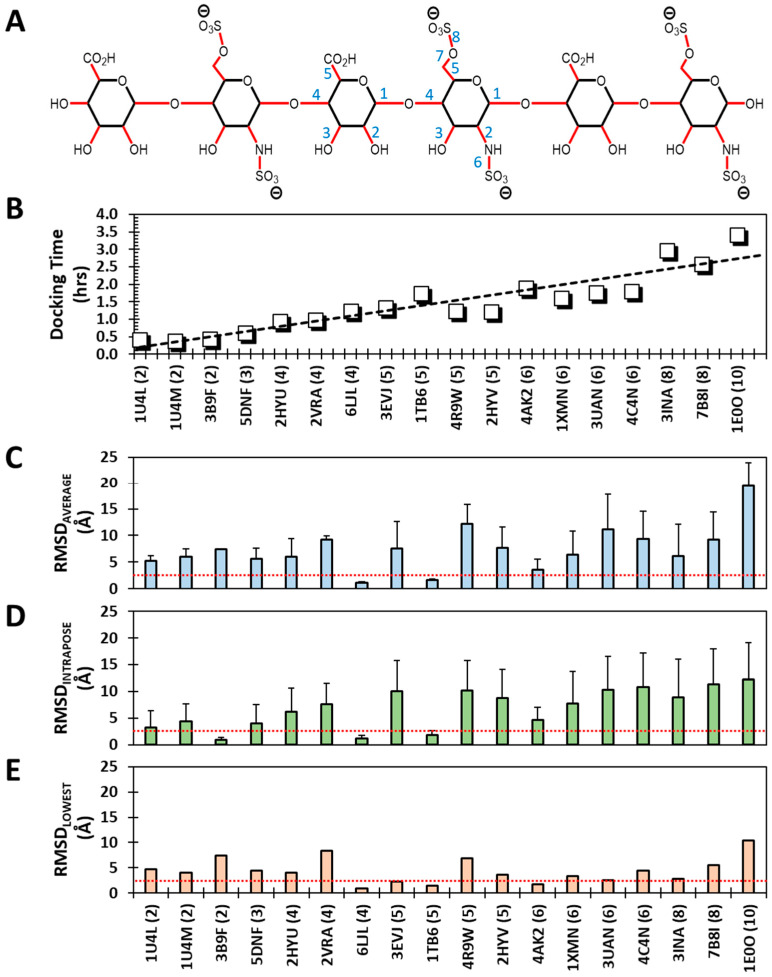
Recapitulation of the native pose using a flexible docking protocol. Each sequence was redocked back into the crystal structure in triplicate using 100 GA runs, each being allowed 100,000 genetic operations. The top two poses from each replicate experiment were selected, compiled and used for analysis. (**A**) The flexible docking protocol affords full flexibility to glycosidic bonds and ring substituents (shown in red). A typical HS hexasaccharide encompasses more than 36 rotatable bonds arising from a minimum of 5 and 7 rotatable bonds in UA and GlcN residues (labeled 1→7 in blue). Ring puckers are held invariant from their starting state in the flexible docking protocol. (**B**) As expected, GOLD dock time for fully flexible docking increased linearly with chain length; although this does not imply that flexible docking yields recapitulation of the native pose (see text for details). (**C**–**E**) Three different RMSDs as function IDs of the co-complex structures reported in the PDB. *X*-axis labels represent the PDB code followed by chain length in brackets. Red dotted line indicates the 2.5 Å cut-off.

**Figure 4 biomolecules-13-01633-f004:**
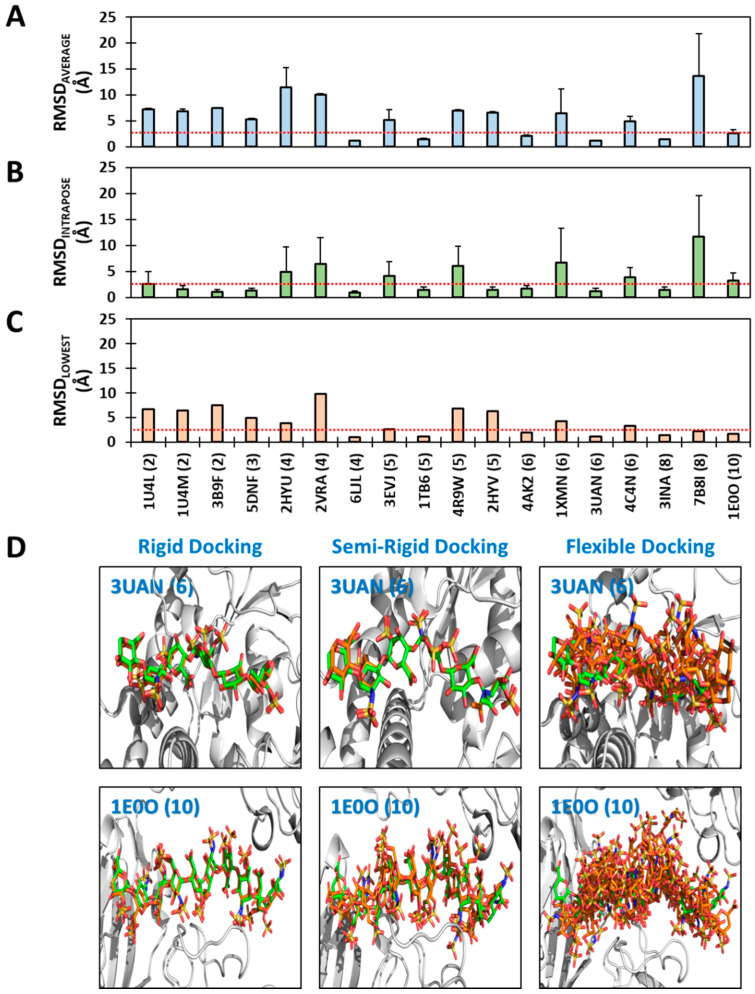
Recapitulation of the native pose using a semi-rigid docking (SRD) protocol. As for rigid and flexible protocols, each sequence was redocked back into the crystal structure in triplicate using 100 GA runs, each being allowed 100,000 genetic operations. (**A**–**C**) Three different RMSDs as function IDs of the co-complex structures reported in the PDB. *X*-axis labels represent the PDB code followed by chain length in brackets. The red dotted line indicates the 2.5 Å cut-off. (**D**) Successful recapitulation of native poses of 3UAN and 1E0O by rigid (left) and semi-rigid (middle) docking protocols but not by the flexible docking protocol (right). Docked poses (shown in orange) are superimposed on native poses (green) for the two sequences. The protein ribbon is shown in light grey.

**Figure 5 biomolecules-13-01633-f005:**
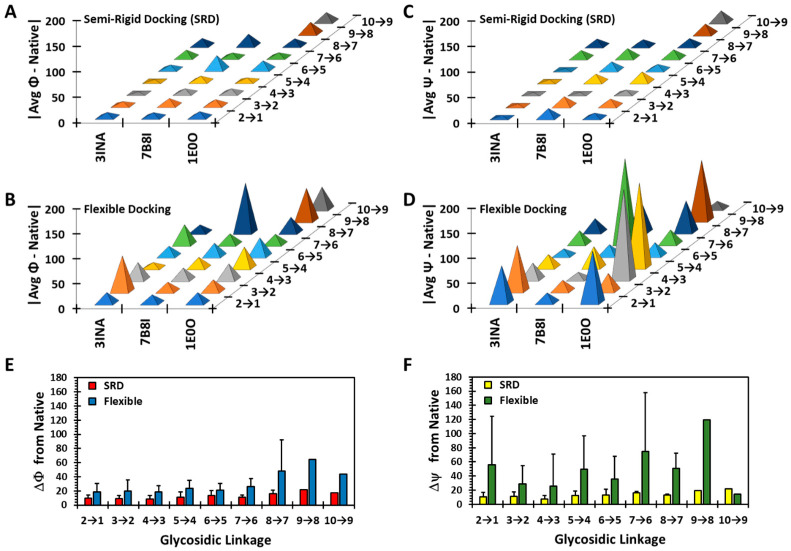
Variation in observed Φ/Ψ from the native pose following semi-rigid and flexible dockings. Shown are Δφ and Δψ, the differences in φ (**A**,**C**,**E**) and ψ (**B**,**D**,**F**) between the native pose and the average of the docked poses obtained following semi-rigid (**A**,**C**) and flexible (**B**,**D**) dockings, respectively, as a function of the co-complex structure and glycosidic bonds (2→1, 3→2, etc., where 2→1 refers to the glycosidic bond between the reducing end residue #1 and the penultimate residue #2). Although the difference (Δφ and Δψ) could be either negative or positive, only the magnitude is shown (i.e., mod of Δφ and Δψ). (**E**,**F**) show the average Δφ and Δψ, respectively, across all 18 sequences from di- to decasaccharide for SRD and flexible docking protocols. See text for details.

**Figure 6 biomolecules-13-01633-f006:**
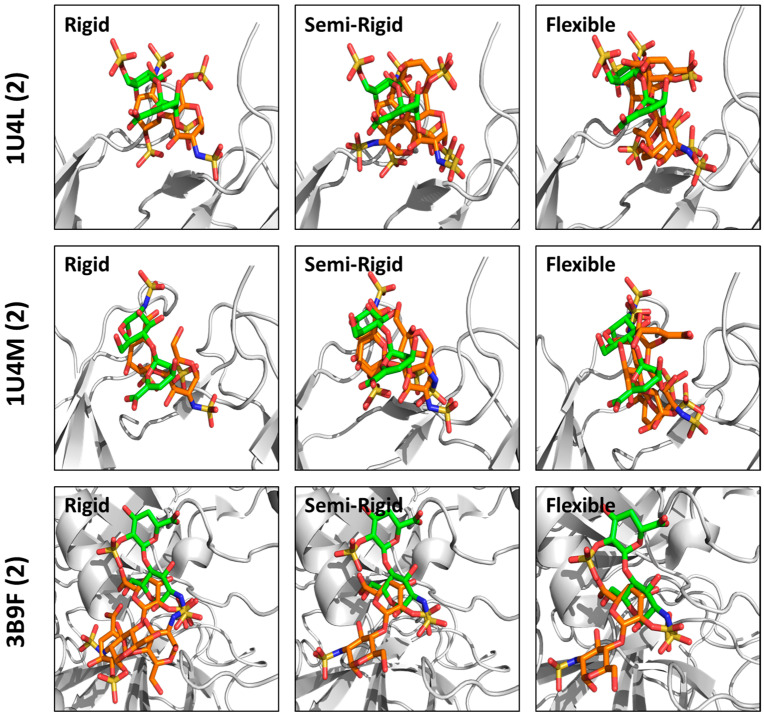
Comparison of docked poses with the native pose of disaccharides. The disaccharide sequence from 1U4L, 1U4M and 3B9F was docked onto the protein in triplicate using 100 GA runs (100,000 genetic operations) using either rigid, semi-rigid or flexible docking protocols. The top two poses from each replicate experiment were selected, compiled and used for visualization. The native pose is shown in green. Docked poses are in orange.

**Figure 7 biomolecules-13-01633-f007:**
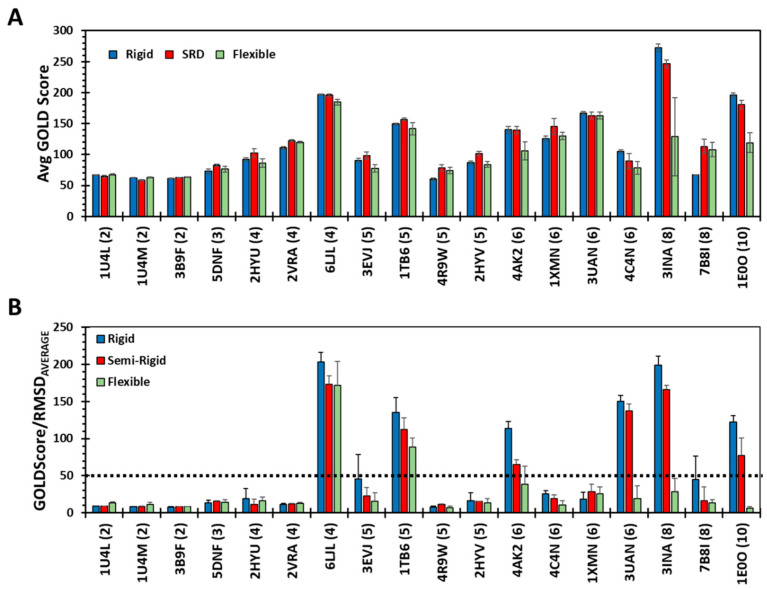
Identifying high-affinity, high-selectivity GAG sequences. (**A**) Average GOLD Scores calculated for six docked poses following rigid, semi-rigid and flexible docking of the 18 Hp/HS sequences onto their target proteins. GOLD Scores reported here were calculated for the protocols with 100 GA runs, each with 100,000 operations. *X*-axis labels represent the PDB code followed by chain length in brackets. Errors show the standard deviation of scores observed in triplicate docking experiments. (**B**) A plot of the ratio of the average GOLD Score to RMSD_AVERAGE_ for rigid, semi-rigid and flexible docking of the 18 Hp/HS sequences. The black dotted line shows an arbitrary cut-off (Ratio ≥ 50) that can be used to identify high-affinity, high-selectivity sequences. From the 18 sequences studied here, only one tetra- (6LJL), one penta- (1TB6), two hexa- (4AK2 and 3UAN), one octa- (3INA) and one decasaccharide (1E0O) are predicted to pass the threshold.

## Data Availability

All raw data for 100 and 300 GA runs including configuration files, torsional distribution files used in SRD, output files (i.e., docked poses, rankings, etc.), native poses of ligands, ligand files and protein files can be requested from authors. These files have also been deposited with the publisher in two compressed files named 100_GA_raw.zip and 300_GA_raw.zip and are available online for free download.

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
