# Peer review of "Assessing Genetic Algorithm-Based Docking Protocols for Prediction of Heparin Oligosaccharide Binding Geometries onto Proteins"

_biomolecules, 2023, doi:10.3390/biom13111633_

Round 1
Reviewer 1 Report
Comments and Suggestions for Authors
In the current manuscript, the authors investigate the binding of glycosaminoglycans (GAGs) to proteins, which is challenging due to the complex nature of GAGs and the lack of computational tools and libraries for such studies. Authors perform a rigorous comparative study on structurally diverse group of proteins exhibiting diversity of GAG recognition selectivities and conclude that the rigid and semi-rigid protocols recapitulate crystal structure poses for longer chains better than the flexible protocol. The authors present a new semi-rigid protocol in combination with new computational parameters that would be particularly useful for GAG-protein docking and provide insights into the selectivity and non-selectivity of these interactions.
The article is well structured into sections and subsections. The introduction is comprehensive and well written. The article is within the scope of the journal. It will be of interest to the readers of the journal.
There is a minor suggestion to improve the article:
1) Page 14, Discussion: Authors are suggested to make Discussion section as a separate heading for better legibility.
Author Response
Thanks for noticing the mistake. We have moved 'Discussion' to a separate heading.
Reviewer 2 Report
Comments and Suggestions for Authors
In the manuscript titled “Assessing Genetic algorithm-based docking protocols for prediction of heparin oligosaccharide binding geometries on to proteins”, Holmes and Desai used genetic-algorithm based docking with rigid, semi-rigid, and flexible docking protocols to study 18 heparin/heparan sulfate-protein complexes. The authors reported that the rigid and semi-rigid docking protocols capture native poses for longer chains better than the flexible docking protocols. However, small chains and the systems with no available crystal structures were better predicted by the semi-rigid docking protocols. They introduced a new parameter based on root mean squared deviations and Gold docking scores to parse selective versus non-selective systems. The work was executed reasonably, and the write-up is organized well and presented in an acceptable manner.
The authors should address the following issues before consideration for the publication.
1. The authors list “SybylX” for the preparation, minimization, and visualization. GOLD was used for molecular docking. How reasonable is the selection of GOLD as the docking program to capture the expected results, especially in the cases where there is no crystal structures were available? How good Gasteiger-Huckel charges in representing charged building blocks of Hp/HS systems? Since the amino acids sidechains of protein were kept fixed, how would it influence the docking results? It is unclear that the cut off +/-30 was employed for selected values of phi and psi of each system or there is a general average for phi and psi.
2. It is not clear how the phi and psi values were fixed in the rigid docking. Specially in the case of systems with unknown pdb structures, what are the selected rigid values of phi and psi.
3. In the present study, quoting the literature, the authors mentioned that a value of 2.5 Ang or less for the root mean square deviation provides a geometric equivalence between the two species. Should this selection depend on the chain length? That means, the dimer and the decamer should not have the same cut off for rmsd for two species to be equivalent. Therefore, is it meaningful to use of such a cut-off (figures 2,3, and 4) to evaluate protocols?
4. In the current study, the authors reported that in the current work, several co-crystal poses were not captured by any of the three docking approaches implemented. Would that simply be a result of using simple charge distributions such as Gasteiger-Huckel charges? Or is it related to the way that GOLD scores the poses? If that is the case, how can one trust the predictions on the systems where there were no available crystal structures based on the scheme implemented in the present study?
Author Response
See attached document

Reviewer 3 Report
Comments and Suggestions for Authors
This manuscript reports the evaluation of new “semi-rigid” docking protocols for analyzing the interactions of glycosaminoglycans (GAGs) with a wide variety of proteins. The authors compare the fidelity of docking interactions observed when using the new semi-rigid protocols as compared to more traditional “flexible” and “rigid” docking protocols by first thoroughly evaluating the rigid and flexible approaches and then comparing these to the semi-rigid approach. The authors thoroughly describe and justify the parameters utilized in their novel semi-rigid docking approach and the cases in which this semi-rigid protocol outperformed the existing protocols. Other key findings such as inconsistencies with the docking of disaccharides and the development of a novel metric for evaluating the “drug-like potential” of GAGs were also reported. The authors effectively describe the importance and utility of this work and provide excellent context in the introduction of this manuscript. The experimental methods are described in sufficient detail. The manuscript is generally exceptionally well-written and clearly worded. A few minor editing errors were identified which should be rectified by a final edit/proofreading of the manuscript. In some cases, the images of figures do not appear on the same page as the figure caption, which should be rectified. This reviewer believes this manuscript would be of significant interest to the readership of biomolecules and should be published.
Author Response
We thank the reviewer for the suggestions. We have taken care of those in the revised manuscript.